# Using mass spectrometry to investigate fluorescent compounds in squirrel fur

**Bryan Hughes** [ID]<sup>1©¤</sup>*, **Jeff Bowman**<sup>2©</sup>, **Naomi L. Stock**<sup>3©</sup>, **Gary Burness**<sup>1©</sup>

**1** Department of Biology, Trent University, Peterborough, Ontario, Canada, **2** Ontario Ministry of Northern Development, Mines, Natural Resources and Forestry, Trent University DNA Building, Peterborough, Canada, **3** Water Quality Centre, Trent University, Peterborough, Ontario, Canada

☉ These authors contributed equally to this work.
¤ Current address: Department of Biology, Laurentian University, Sudbury, Ontario, Canada
* Bhughes1@laurentian.ca

**Data Availability Statement:** All relevant data are within the paper and its supporting information files.

**Funding:** This research is funded by the natural sciences and engineering research council

## Abstract

While an array of taxa are capable of producing fluorescent pigments, fluorescence in mammals is a novel and poorly understood phenomenon. A first step towards understanding the potential adaptive functions of fluorescence in mammals is to develop an understanding of fluorescent compounds, or fluorophores, that are present in fluorescent tissue. Here we use Fourier transform-ion cyclotron resonance mass spectrometry (FT-ICR MS) of flying squirrel fur known to fluoresce under ultraviolet (UV) light to identify potentially fluorescent compounds in squirrel fur. All of the potentially fluorescent compounds we identified were either present in non-fluorescent fur or were not present in all species of fluorescent flying squirrel. Therefore, we suggest that the compounds responsible for fluorescence in flying squirrels may also be present in non-fluorescent mammal fur. Some currently unexplained factor likely leads to excitation of fluorophores in flying squirrel fur. A recently suggested hypothesis that fluorescence in mammals is widely caused by porphyrins is consistent with our findings.

## Introduction

Ultraviolet (UV) fluorescence is a physicochemical phenomenon present in an array of biological taxa and geographical landscapes [1]. There is substantial research on the occurrence of fluorescence in plants [2] and invertebrates [3,4]. There is also increasing evidence to suggest that many vertebrates have fluorescent pigments, including birds [5–7], amphibians [8], reptiles [9,10], fish [11] and mammals [1,12–15]. Several hypotheses aim to explain an ecological role of fluorescence in vertebrates. For example, it has been suggested that fluorescence may function as an anti-predator defence mechanism, such as camouflage [11], aposematism [16] or mimicry [1]. Alternatively, some species may use fluorescence as a visual signal for mating [7,17,18]. In many cases, the role of fluorescence is either not fully understood [19,20], the result of an artefact, or lacks sufficient evidence to suggest a specific function [21]. Further investigation into the chemical mechanisms resulting in fluorescence may provide insight into the potential ecological role of fluorescence in vertebrates.

discovery grant held by JB. The funders had no role in study design, data collection and analysis, decision to publish, or preparation of the manuscript.

**Competing interests:** The authors have declared that no competing interest exist.

Fluorescence occurs when a molecule, known as a fluorophore, is excited by electromagnetic radiation. Fluorescence is not the formation of light, and therefore, fluorophores do not create a source of light, glow, or other form of energy. Rather, this phenomenon is a form of luminescence in which a molecule emits light that is absorbed from an external source. The excitation of a fluorophore causes electrons within the molecule to move to an energized state. As the electron transitions back to its ground state, excess energy is lost in the form of a photon [22,23]. The energy emitted from this reaction produces the vivid fluorescent colours seen in many different ecological systems. Since fluorescence is not the only method for a molecule to return to its ground state, there are a few structural characteristics that can be used to predict if a molecule can fluoresce. Generally, fluorescence requires a planar conjugated system with alternating single and double carbon bonds. For example, aromatic molecules such as benzene are capable of fluorescence [24,25]. Other molecules that contain an aromatic ring, such as tryptophan and its derivatives are also capable of fluorescence at various wavelengths [13]. The energy of a photon dictates the fluorescence and perceived wavelength of a fluorophore. The energy is dependent on the de-excitation of an electron from the lowest unoccupied molecular orbital (LUMO) to the highest occupied molecular orbital (HOMO) [26,27]. Aromatic rings with highly conjugated systems have comparatively small HOMO-LUMO energy gaps that readily absorb electromagnetic radiation and emit fluorescence [27,28]. In contrast, non-aromatic or non-conjugated molecules may have a large energy gap, that is less likely to absorb ultraviolet light and produce fluorescence [26]. Therefore, any compound meeting these characteristics should be potentially fluorescent.

At one time, only aromatic compounds were thought to be fluorescent, however non-aromatic compounds that contain conjugated π-bonded systems (systems that contain one or more covalent double or triple bonds), may also fluoresce [23,29,30]. Fluorescent compounds that lack an aromatic ring include non-aromatic steroid hormones such as cortisone and aldosterone [29].

Across taxa, several different fluorophores have been identified. For example, many species of parrot have fluorescent plumage due to specialized psittacofulvine pigments that fluoresce in an array of colours [31]. The plumage of many nocturnal birds, such as owls (Strigiformes) and nightjars (Caprimulgiformes) fluoresce pink under a 395 nm blacklight, due to an accumulation of the compound coproporphyrin III [5]. Fluorophores in amphibians may include dihydrosoquinolinone derivatives, such as those found in the glandular secretions of certain tree frogs [8]. The fluorophores in tree frogs appear to fluoresce blue or green when exposed to a 400 nm UV-blue light [8]. In fish, there are a variety of different compounds and proteins responsible for different fluorescent patterns and colours [11,32]. An accumulation of uroporphyrin I causes pink fluorescence in the teeth and bones of eastern fox squirrels (*Sciurus niger*) [33]. Although there are many known fluorophores, the specific cause of fluorescence in many species remains unknown. For example, the shells of certain marine turtles [9] and the beaks of puffins (*Fratercula* spp.) [6] are known to fluoresce (green and blue, respectively), however the chemical mechanism causing this is unclear. Identifying potential fluorophores is a first step in understanding whether fluorescence has a function, and what that function may be.

Historic observations suggested that fluorescence in mammals was limited to the least weasel (*Mustela nivalis*) and some Australian marsupials such as the red kangaroo (*Macropus rufus*) and the grey possum (*Trichosurus vulpecula*) [34,35]. More recent research found no evidence of fluorescence within the least weasel or red kangaroo and therefore fluorescence was believed at one time to be limited to Didelphidae [13,36]. Fluorescence has now been observed in all major taxonomic clades of mammals. Within eutherians, fluorescence has been observed in the fur of all extant species of North American flying squirrel (*Glaucomys* spp.) [1], nocturnal springhares (*Pedetidae* spp.) [12], the Coxxings white-bellied rat (*Niviventer*

*coninga*), the scales of the Chinese pangolin (*Manis pentadactyla)* [14], and the quills of European hedgehogs (*Erinaceus europaeus*) [37]. The fur of monotremes such as the platypus (*Ornithorhynchus anatinus*) is also fluorescent [15]. Fluorescence in the fur of marsupials has also recently been shown to include wombats (*Diprotodontia* spp.), gliders (*Petaurus* spp.), and bandicoots (*Peramelidae* spp.) [38]. There is evidence to suggest that some nocturnal species may be capable of detecting ultraviolet light [39,40]. For example, the density of photoreceptor cells (opsins) in relation to the structure of the eye in the Ord's Kangaroo rat (*Dipodomys ordii*) suggests that this species is capable of UV detection [39]. More generally, most diurnal species possess a yellow eye-lens that filters UV light protecting the retina. This yellow eye-lens is absent in many nocturnal species, including North American flying squirrels, where a clear eye lens may be indicative of an increased ability to absorb UV light in low-light environments [40]. Most recently, Toussaint et al. (2021) [41] have suggested that fluorescence in mammals is widespread and caused by an accumulation of porphyrins. While all mammals should contain porphyrin compounds, they further suggested that porphyrins break down through exposure to UV, and consequently, we might expect fluorescence to be more common among nocturnal species. This is one of many possible hypotheses that attempt to explain fluorescence in mammals.

While porphyrins appear to cause fluorescence in the bones, teeth, and pelage of certain mammals [32,41], other fluorophores are known to cause fluorescence in mammal fur, including tryptophan and tryptophan derivatives [13]. Tryptophan is one of three natural aromatic amino acids [24], while tryptophan derivatives, including 3-hydroxlanthanalic acid, kynurenine, kynurenic acid, and xanthurenic acid are all conjugated molecular systems that contain an aromatic ring. While the presence of an aromatic ring does suggest that these compounds are capable of fluorescence, they exhibit a range of different fluorescent colours. The colour of fluorescence of a compound is governed by the position of different functional groups [42]. For example, possums exhibit a range of fluorescent colours dependent on the stage of tryptophan catabolism [13]. The compound 3-hydroxyanthalinic acid is a product of tryptophan catabolism that produces a blue fluorescence [13]. Similarly, tryptophan derivatives such as kynurenic acid and xanthurenic acid are known to produce red fluorescence, while kynurenine produces a yellow fluorescence [13].

Fluorescence in eutherian mammals is a poorly understood phenomenon. The dorsal pelage of North American flying squirrels has a blue, fluorescent hue, whereas the ventral pelage is a more vibrant pink (Fig 1). There is no apparent difference in intensity between sex, habitat, or locality [1]. Due to the extensive geographic range of flying squirrels, it has been suggested as unlikely that the observed fluorescence is the product of diet [1]. We currently do not understand the cause of fluorescence in flying squirrels, or whether this is an adaptive trait. Several potential adaptive benefits of fluorescence in flying squirrels may include anti-predator defense mechanisms and social communication; however, fluorescence may also be an artefact [1,41]. We consider establishing biochemical pathways a good starting point to better our understanding of the evolutionary processes leading to this trait.

Here, we use mass spectrometry to investigate potentially fluorescent compounds within the fur of North American flying squirrels. We hypothesize that their fur should contain potentially fluorescent compounds that do not occur in fur from non-fluorescing squirrel species.

## Methods and materials

### Preparation of fur samples

No live animals were used in this study. All fur samples were obtained from archived specimens from a previous study that were collected following a protocol approved by the Trent

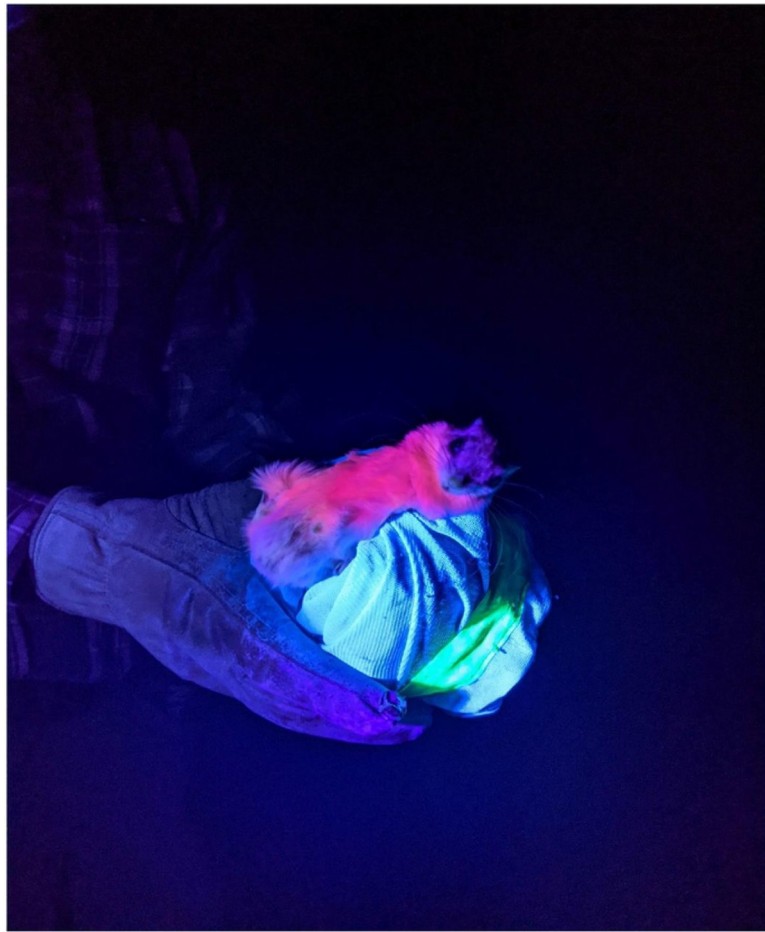

**Fig 1. Fluorescent flying squirrel.** A male southern flying squirrel (*Glaucomys volans*) being held in a bag fluorescing under a UV blacklight. Showing the ventral portion of the body with the head on the Right. Photo courtesy of Rebekah Persad, Trent University, Peterborough Ontario.

University animal care committee (Protocol 25668) [43]. We used fur samples from two known fluorescing squirrel species and three non-fluorescing species (Table 1). All samples were taken from frozen specimens collected with authorization. Fluorescent samples included one northern and one southern flying squirrel (*Glaucomys sabrinus* and G. *volans* respectively). Two ventral samples, and one dorsal sample were obtained from each of the two

**Table 1. The total number of ventral and dorsal samples retrieved from each individual specimen, showing both species examined for fluorescent fur, and the three species used to compare non-fluorescent fur determined using a 395 nm blacklight.**

| Specimen number | Common name | Species name | Ventral samples taken | Dorsal samples taken | Fluorescent (Y/N) |
|---|---|---|---|---|---|
| 1 | Northern flying squirrel | *Glaucomys sabrinus* | 2 | 1 | Yes |
| 2 | Southern flying squirrel | *Glaucomys volans* | 2 | 1 | Yes |
| 3 | Eastern chipmunk | *Tamias striatus* | 2 | 0 | No |
| 4 | Gray squirrel | *Sciurius carolinensis* | 1 | 1 | No |
| 5 | Gray squirrel | *Sciurius carolinensis* | 1 | 1 | No |
| 6 | Red squirrel | *Tamiasciurius hudsonicus* | 1 | 1 | No |
| 7 | Red squirrel | *Tamiasciurius hudsonicus* | 1 | 1 | No |

fluorescing species. For each sample, approximately 20 mg of fur were collected by trimming the squirrel hair, and not taking the hair follicle.

To obtain fur from non-fluorescing species, we took fur from two gray squirrels (*Sciurius carolinensis*), two red squirrels (*Tamiasciurus hudsonicus*) and one eastern chipmunk (*Tamias striatus*). Approximately 20 mg of fur from the ventral pelage of the two red squirrels; approximately 40 mg of fur from the ventral pelage of the chipmunk, split into two 20 mg samples; and 20 mg of fur from the dorsal region of each grey squirrel was collected.

All fur samples were placed into separate 50 mL polypropylene centrifuge tubes filled with 5 mL of methanol (HPLC grade; Thermo Fisher Scientific, Whitby ON Canada). To help isolate ions for the mass spectra analysis, samples were sonicated (Model 8892; Cole-Parmer, Barrie ON Canada) for 10 minutes Finally, we placed each sample in a centrifuge (Model IEC, CentraCL2; Thermo Fisher Scientific) at 4000 rpm for 4 minutes to help separate the methanol from the solid fur. We used four centrifuge tubes filled with only 5 mL of methanol as control samples to account for any ions present due to the methanol or centrifuge tubes.

## Mass spectrometry analysis

We used mass spectrometry analysis to test for the presence of potentially fluorescent compounds in squirrel fur. Methanolic extracts were analyzed using a Bruker SolariX XR Fourier transform-ion cyclotron resonance mass spectrometer (FT-ICR-MS) (Billerica MA USA) equipped with a 7T magnet and an electrospray ionization (ESI) source and located in the Water Quality Centre at Trent University. Prior to analyses, the system was externally mass calibrated using sodium trifluoroacetate, 0.1 mg/mL in methanol. All samples were analyzed in both positive and negative ion modes over a mass range of *m/z* 54 to 2000. Spectra were acquired using Bruker ftms Control software (version 2.1.0). For each sample, 100 scans were obtained, with 1 million data points collected per scan. The free ion decay (FID) was 0.2621 s. The temperature of the ESI source was 200˚C, nebulizer gas was set to 1 bar, and dry gas flow rate was 4 L/min. Samples were infused into the ESI source using a flow rate of 180 μL/hr.

## Fluorescence analysis

Aliquots (2 ml) of each methanolic extract were concentrated 10-fold using a nitrogen evaporator (Organomation; Berlin MA, USA) and high purity nitrogen (Praxair Peterborough ON, Canada) and analyzed for fluorescent intensity at different wavelengths using a SpectraMax M3. Multi-Mode Microplate reader (Molecular Devices; San Jose, CA, USA). For each sample, 1 ml was extracted using a 1 ml Eppendorf pipette and placed into a polystyrene cuvette for analysis. Associated curves were obtained using SoftMax Pro at a fixed excitation wavelength of 350 nm, scanning for emissions at 400 nm to 600 nm. These parameters were selected based on previous knowledge of fluorescence in flying squirrels, where all specimens appear to elucidate and absorb wavelengths of at least 395 nm and emit pink (ventral) and blue (dorsal) colours [1].

## Data analysis

Mass spectra were analyzed using the Bruker Compass Data Analysis software (Version 5.0) and a peak list was obtained for each sample. Each peak corresponds to a unique mass-to-charge value (*m/z*). To isolate potential fluorophores, we performed two differential analysis tests of our mass spectral data. First, we evaluated known fluorescing and non-fluorescing squirrel fur, in both positive and negative ion modes. The fluorescent samples included both dorsal and both ventral samples from each of the two species of flying squirrel. Peaks that were observed in both fluorescing and non-fluorescing fur samples were excluded, resulting in a list

of potential ions of interest. For each ion of interest, a tentative chemical formula and mass accuracy was determined using the Bruker Compass Data Analysis software and Metlin [44]. Mass accuracy was calculated using:

$$\text{mass accuracy (ppm)} = \frac{M_{observed} - M_{calculated}}{M_{calculated}} \times 10^6$$

where $M_{observed}$ is the experimental $m/z$ value and $M_{calculated}$ is the calculated $m/z$ value. Values may be positive, indicating that the observed $m/z$ is larger than the calculated $m/z$, or negative, indicating that the observed $m/z$ is smaller than the calculated $m/z$.

Second, we examined the peak lists, obtained in both positive and negative ion modes, from ventral and dorsal fur samples of fluorescent fur, irrespective of flying squirrel species. Due to the different colour hues found in the different pelage locations on flying squirrels, all fur from the ventral pink portion of the northern and southern flying squirrel were examined separately from the dorsal blue fur. We recorded all peaks found only in the ventral or dorsal fur samples; peaks that were also observed in the fur of the non-fluorescing species were excluded. Again, we used Bruker Compass Data Analysis software to determine a tentative chemical formula and mass accuracy for each ion of interest.

## Results

### Analysis of ions of interest in fluorescent and non-fluorescent fur

Following FT-ICR MS analysis, we compiled a list of positive (Table 1) and negative (Table 2) ions observed only in fluorescent flying squirrel fur samples. Several of these compounds may have the necessary chemical structure to indicate capability to fluoresce. However, we did not identify any compounds present in both flying squirrel species that were absent in the three non-fluorescing species, which may have been indicative of the compound responsible for the fluorescence in flying squirrels. Overall, we identified 20 tentative compounds that may be capable of fluorescence. While our initial analysis did not find any compounds present in both fluorescing species, we found eleven positive ions (Table 2) and four negative ions (Table 3) present in one of the fluorescing species but none of the non-fluorescing species.

Four of the positive ions identified in fluorescent fur occurred in the southern flying squirrel samples, while the remaining seven positive ions were found in the northern flying squirrel samples. All four negative ions found, were present in one or more of the northern flying squirrel samples, but not in any of the southern flying squirrel samples.

**Table 2. Positive ions found in all examined southern (*Glaucomys volans*) or northern (*Glaucomys sabrinus*) flying squirrel fur samples, with tentative chemical formula.**

| Species | Mass observed (m/z) | Potential formula (M) | Ion Observed | Mass calculated (m/z) | Mass accuracy (ppm) |
|---|---|---|---|---|---|
| *Glaucomys volans* | 152.13025 | $C_9H_{15}N_2$ | $[M+H]^+$ | 152.1308 | -3.61 |
| *Glaucomys volans* | 297.03567 | $C_{11}H_8N_2O_8$ | $[M+H]^+$ | 297.0353 | 1.25 |
| *Glaucomys volans* | 333.29533 | $C_{19}H_{41}O_2P$ | $[M+H]^+$ | 333.2917 | 10.89 |
| *Glaucomys sabrinus* | 142.06309 | $C_5H_7N_3O_2$ | $[M+H]^+$ | 142.0611 | 14.01 |
| *Glaucomys sabrinus* | 194.19035 | $C_{13}H_{23}N$ | $[M+H]^+$ | 194.1903 | 0.26 |
| *Glaucomys sabrinus* | 201.06423 | $C_8H_{10}NO_5$ | $[M+H]^+$ | 201.0632 | 5.22 |
| *Glaucomys sabrinus* | 216.07481 | $C_8H_{11}N_2O_5$ | $[M+H]^+$ | 216.0741 | 3.42 |
| *Glaucomys sabrinus* | 309.25849 | $C_{23}H_{32}$ | $[M+H]^+$ | 309.2577 | 2.55 |
| *Glaucomys sabrinus* | 323.29311 | $C_{19}H_{40}O_2$ | $[M+Na]^+$ | 323.2920 | 3.43 |
| *Glaucomys sabrinus* | 349.21466 | $C_{22}H_{26}N_3O$ | $[M+H]^+$ | 349.2149 | -0.57 |
| *Glaucomys sabrinus* | 349.25123 | $C_{18}H_{37}O_4P$ | $[M+H]^+$ | 349.2506 | 1.72 |

**Table 3. Negative ions observed in either the northern (*Glaucomys sabrinus*) or southern (*Glaucomys volans*) flying squirrel fur, with tentative chemical formula.**

| Species | Mass observed (m/z) | Potential formula (M) | Ion Observed | Mass calculated (m/z) | Mass accuracy (ppm) |
|---|---|---|---|---|---|
| *Glaucomys volans* | 85.06635 | $C_5H_{10}O$ | [M-H]⁻ | 85.0658 | 5.41 |
| *Glaucomys volans* | 155.14434 | $C_{10}H_{20}O$ | [M-H]⁻ | 155.1441 | 1.29 |
| *Glaucomys volans* | 251.23845 | $C_{17}H_{32}O$ | [M-H]⁻ | 251.2380 | 1.63 |
| *Glaucomys volans* | 291.23335 | $C_{19}H_{32}O_2$ | [M-H]⁻ | 291.2330 | 1.20 |

## Analysis of ions of interest within either ventral or dorsal fluorescent pelage

We also found five compounds during our comparison of ventral and dorsal pelage, irrespective of flying squirrel species (Table 4). Of the five compounds, three compounds were found only in dorsal pelage, where two were found only in ventral pelage. These ions may contribute to the different colours of fluorescence observed in the dorsal and ventral fur of flying squirrels.

## Fluorescence analysis

Our analysis of the fluorescent properties of all squirrel samples (S1 File) showed clear fluorescent emission when excited at 350 nm in flying squirrels compared to our blank samples filled with methanol. However, the samples containing gray and red squirrel fur both showed some capability of fluorescence when excited with the same 350 nm wavelength. The excitation of all samples at 350 nm may be indicative of potential fluorophores being present in both fluorescing and non-fluorescing fur.

## Discussion

While fluorescence occurs in an array of species, the evolution and function of fluorescence in mammals is poorly understood. Before we can understand the potential for any function of fluorescence in mammals, we believe it is helpful to understand the identity and function of mammalian fluorophores. We hypothesised that flying squirrels with fluorescent fur would have a unique compound present within their fur that is not present within non-fluorescent fur of closely related squirrel species. To determine potentially fluorescent compounds, we performed two different mass spectral data analyses. We observed unique compounds present in the fluorescent fur of northern and southern flying squirrels that were not present in the fur of red squirrels, grey squirrels, or chipmunks. We also observed compounds present in either the dorsal or ventral fur irrespective of the flying squirrel species. However, we did not find a single unique compound present in both flying squirrel species that was not also present in the other squirrel species. Examining the fluorescence of all squirrel samples also showed that all

**Table 4. Unique ions found in only dorsal or only ventral pelage of northern (*Glaucomys sabrinus*) and southern (*Glaucomys volans*) flying squirrel species.** Samples from the two species were pooled for analysis.

| Pelage | Mass observed (m/z) | Potential formula (M) | Ion Observed | Mass calculated (m/z) | Mass accuracy (ppm) |
|---|---|---|---|---|---|
| Dorsal | 154.12374 | $C_9H_{17}NO$ | [M-H]⁻ | 154.12374 | 1.75 |
| Dorsal | 217.18099 | $C_{12}H_{26}O_3$ | [M-H]⁻ | 217.18092 | -0.32 |
| Dorsal | 231.17562 | $C_{16}H_{24}O$ | [M-H]⁻ | 231.17544 | -0.78 |
| Ventral | 443.20850 | $C_{26}H_{28}N_4O_3$ | [M-H]⁻ | 443.20886 | 0.81 |
| Ventral | 580.50472 | $C_{34}H_{67}N_3O_4$ | [M-H]⁻ | 580.50588 | 1.99 |

fur contained some compounds that were capable of fluorescing. Therefore, we suggest that the compound responsible for the vivid pink and blue fluorescence in flying squirrels, is likely present in other, non-fluorescing sciurids. Since the fur of other squirrel species does not naturally fluoresce when exposed to a 395 nm blacklight, it is likely that another factor is responsible for the excitation or production of fluorophores in flying squirrels. Fluorescence in nocturnal mammals may therefore rely on the accumulation of a certain fluorophore, an additional biochemical pathway, or some currently unknown factor such the degradation of fluorophores in diurnal mammals [41].

Our expectation to find a unique fluorophore in flying squirrels was based on the presence of unique fluorophores in other taxonomic groups. For example, the skin of some tree frogs is known to produce vivid green, fluorescent pigments due to the presence of dihydroisoquinolinone derivatives, or hyloins [8]. Similarly, a unique group of pigment-compounds known as psittacofulvines are responsible for the bright plumage, and fluorescence of parrot feathers [31,45,46]. Unlike the unique fluorophores found in tree frogs and birds, all potentially fluorescent compounds that we observed in flying squirrel fur were either present in non-fluorescent fur from other squirrel species or were not found in samples taken from both the dorsal and ventral regions of the flying squirrels (both of which fluoresce). Therefore, the specific fluorophore responsible for fluorescence in flying squirrels may be a compound present in non-fluorescent sciurid, or even all mammalian fur.

Because we did not find a unique fluorophore present in flying squirrel fur at the excitation and emission frequencies employed, we did not find support for our initial hypothesis that flying squirrel fur would have unique fluorescent compounds not present in other squirrel species. Instead, we consider it possible that the fluorophore responsible for fluorescence in flying squirrels is common across all of the squirrel species we studied. Our results suggest that (1) there is no unique compound being produced or accumulated in flying squirrels that would not be found in other squirrel species; and (2) there are several fluorophores found in squirrel fur that may not always lead to observable fluorescence. While our results are inconclusive in answering why some species are fluorescent whereas others are not, our findings are consistent with the suggestion of Toussaint et al. (2021) that fluorophores may be more widespread in mammal fur than previously understood [40]. Toussaint et al. (2021) suggested that fluorescence in mammals is the result of porphyrins accumulated in tissue through the heme pathway and proposed that fluorescence is more common in nocturnal species because porphyrins are photodegradable [40,47]. This hypothesis suggests that fluorescence in mammals is only more apparent in nocturnal species than diurnal species because of photodegradation of porphyrins occurring in diurnal species. If this phenomenon is true, then we would expect that fluorescence of flying squirrel study skins would be reduced if exposed to UV light. While this hypothesis may explain why only certain species are fluorescent, it also suggests that most nocturnal mammal species should be fluorescent. Patterns of photodegradation can be specific to different porphyrin compounds however, and porphyrins may accumulate in tissue differently based on the compound and specimen [47,48]. Therefore, the fluorescent patterns may also be species-specific regardless of nocturnal cycle. If fluorescence in flying squirrels and some other nocturnal mammals is observable because porphyrins accumulate in tissue at higher rates than other species, it remains possible that there is another mechanism causing the abnormal production of porphyrins. The parameters used to measure fluorescence in this study were not selected to identify porphyrins, therefore, to further test this hypothesis, we would suggest studies focus on evaluating the biochemical synthesis of porphyrins when evaluating the potential role of fluorescence in flying squirrels and other mammals.

Other potential fluorophores in mammal fur are tryptophan derivatives found in possums [13,36]. All possum fur appears to be fluorescent, where different fluorescent hues are

associated with different tryptophan derivatives [13]. Tryptophan and tryptophan metabolites are used within the mammalian body system to produce many hormones. Tryptophan and tryptophan metabolites typically accumulate within the hair bulb of most mammal fur [13]. While tryptophan may be responsible for the fluorescence observed in flying squirrels, fur samples from all species contained tryptophan (S2 and S3 Files); therefore, we were unable to isolate tryptophan as the specific compound responsible for fluorescence in flying squirrels. However, as all mammalian fur with tryptophan is not fluorescent, there may be another physiological mechanism or process responsible for fluorescence. Pine et al. (1985) [13] suggested that the difference in enzymic complement of skin tissue may result in the partial or alternate completion of tryptophan metabolism. As a result, different tryptophan derivatives tend to accumulate within different portions of the possum, resulting in different fluorescent pigments. More research is necessary to understand if the fluorescence of possums is the result of a higher accumulation of tryptophan, or another physiological mechanism altogether.

Perhaps the most intriguing aspect of the recent observations of fluorescent mammals is the potential for fluorescent fur to be an adaptive trait. While it remains uncertain, evidence to date suggests that fluorescence in mammals may often be an artefactual result of biochemical synthesis [41,49]. Fluorescence does appear to be an adaptive trait in some taxa, however. For example, the fluorescent plumage in parrots is believed to enhance visual mating signals, and there is correlation between fluorescent intensity and reproductive success [7,46,50,51]. Many marine fishes fluoresce as a form of camouflage, as they fluoresce in similar colours to the corals in their habitat [11,51]. In many cases, the specific ecological role of fluorescence is simply unknown [9] or may simply be the by-product of pigment producing molecules, or even a retained vestigial trait with no adaptive function [19,21,52]. There are several hypotheses that aim to explain a potential ecological role of fluorescence in mammals [1,41]. First, it is necessary that ultraviolet light be present in a normal environment for the species, and an individual must be able to detect ultraviolet emission [21]. We would also expect that there would be some variation among individuals within the population, and this variation may provide some ecological advantage. If there is a unique fluorophore in flying squirrel fur, it seems that there should be a benefit that outweighs the costs of production of this compound. In contrast, our finding that both fluorescing and non-fluorescing fur may contain potential fluorophores seems to weaken the argument for a potential ecological function of these compounds. More research will be required to clarify whether fluorescence in flying squirrels plays any adaptive role or is artefactual.

In summary, using the sample extraction described and FTICR-MS we did not find a unique fluorophore that appears to be responsible for the vivid fluorescence seen in flying squirrels. We have identified several compounds that may have the characteristics to be fluorescent, but none of these occurred in both flying squirrel species and consequently are not likely the cause of fluorescence in this clade. Our observations suggest that fluorescence in mammal fur is not limited to the observable fluorescence seen in flying squirrels and other nocturnal species. We propose three plausible explanations for our results. First, as proposed by Toussaint et al (2021), fluorescence in flying squirrels and other nocturnal mammals may be the result of porphyrins that photodegrade when exposed to UV light. Second, fluorescence in mammal fur may require a secondary form of activation or reliance on the accumulation of a specific compound common in mammal fur. There is also growing evidence to suggest that fluorescence in mammals may be the result of either porphyrin or tryptophan synthesis [12,13,37,42,49]. Since we did not find a unique fluorescent compound in flying squirrel fur, it remains possible that fluorescence in flying squirrels is the result of some biochemical pathway involving synthesis of one of these two compounds. Because our analysis suggests that fur from both fluorescent, and non-fluorescent species contained some quantity of fluorophore, it

is possible that fluorescence in flying squirrels is caused by the accumulation of a particular common compound or may have an additional physiological component. If this were true, we would expect there to be some variation in fluorescent intensity across individuals within a population. We may also expect that some species may be fluorescent, and others would not be because the production and accumulation of certain compounds is often species specific [13,41]. Finally, it is also possible that the compound responsible for fluorescence was not extracted using methanol-sonication, or not observed using ESI mass spectrometry.

Our research also demonstrates that the fur of mammals known to fluoresce under 395 nm, and fur with no observed fluorescence under the same conditions both contain potential fluorophores. Further research into specific physiological mechanisms in flying squirrels may help provide insight into what is causing fluorescence.

## Supporting information

**S1 File. Fluorescence analysis of fur samples.** The fluorescence analysis of all fur samples observed at an excitation length of 400 to 600 nm, excited at a fixed emissions peak of 350 nm.
(PDF)

**S2 File. Negative ion mode mass spectrometry data.** The mass spectra values for all fur samples using negative ion mode.
(XLSX)

**S3 File. Positive ion mode mass spectrometry data.** The mass spectra values for all fur samples using positive ion mode.
(XLSX)

## Acknowledgments

We would like to thank Phillip Biko Siambi at Noblegen, Peterborough ON, for their help with fluorescence analysis. We would also like to thank lab colleagues, including Sasha Newar, Rebecca Persad, Ryan Holt and many others for support and inspiration throughout this project.

## Author Contributions

**Conceptualization:** Bryan Hughes, Jeff Bowman.

**Data curation:** Bryan Hughes.

**Formal analysis:** Bryan Hughes.

**Funding acquisition:** Jeff Bowman.

**Investigation:** Bryan Hughes.

**Methodology:** Bryan Hughes, Jeff Bowman, Naomi L. Stock.

**Project administration:** Jeff Bowman.

**Writing – original draft:** Bryan Hughes.

**Writing – review & editing:** Bryan Hughes, Jeff Bowman, Naomi L. Stock, Gary Burness.

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
