## [Decision Letter · Decision Letter 0]

14 Oct 2021

PONE-D-21-27099Using mass Spectrometry to investigate fluorescent compounds in squirrel furPLOS ONE

Dear Dr. Hughes,

Thank you for submitting your manuscript to PLOS ONE. After careful consideration, we feel that it has merit but does not fully meet PLOS ONE’s publication criteria as it currently stands. Therefore, we invite you to submit a revised version of the manuscript that addresses the points raised during the review process.

Both reviewers and I found the paper interesting. While reviewer 2 was quite positive about the paper, they also requested some clarifications. Reviewer 1 had substantial methodological concerns, specifically whether the wavelengths you used were suitable for detection of porphyrins. Please pay particular attention to these concerns in your response, as they are critical to the validity of your findings. 

We look forward to receiving your revised manuscript.

Kind regards,

Matthew Shawkey

Academic Editor

PLOS ONE

Journal Requirements:

2. In your Methods section, please provide additional details regarding the animals used in your study and ensure you have described the source. For more information regarding PLOS' policy on materials sharing and reporting, see https://journals.plos.org/plosone/s/materials-and-software-sharing#loc-sharing-materials.

 “This study was funded by the natural sciences and engineering research council discovery grant, JB.”

 “The authors are grateful to the Canadian Foundation for Innovation, and the Ontario Research Fund for funding instrumentation, including the Bruker SolariX XR FT-ICR mass spectrometer, in the Water Quality Centre of Trent University”

  “This study was funded by the natural sciences and engineering research council discovery grant, JB.”

5. Please ensure that you refer to Figure 1 in your text as, if accepted, production will need this reference to link the reader to the figure.

Reviewers' comments:

Reviewer's Responses to Questions

**Comments to the Author**

1. Is the manuscript technically sound, and do the data support the conclusions?

Reviewer #1: No

Reviewer #2: Yes

2. Has the statistical analysis been performed appropriately and rigorously? 

Reviewer #1: N/A

Reviewer #2: Yes

3. Have the authors made all data underlying the findings in their manuscript fully available?

Reviewer #1: Yes

Reviewer #2: Yes

4. Is the manuscript presented in an intelligible fashion and written in standard English?

Reviewer #1: Yes

Reviewer #2: Yes

5. Review Comments to the Author

Reviewer #1: This manuscript by Hughes et al. describes the use of Fourier transform-ion cyclotron resonance mass spectrometry (FT-ICR MS) to identify mass features present in the fluorescence pelage of either the Southern or Northern flying squirrels, but not in the non-fluorescent fur of gray squirrels, red squirrels, or eastern chipmunks. By comparing the spectrometric profiles of fluorescent vs non-fluorescent fur, the authors identified 22 tentative compounds exclusive to at least one of the fluorescent samples; however, none of the compounds were present in both flying squirrel species. Moreover, through fluorescence spectroscopy, the authors propose that both the fluorescent squirrels and some of the non-fluorescent animals exhibit fluorescence. These findings lead the authors to claim that compounds responsible for fluorescence in flying squirrels may also be present in non-fluorescent mammal fur, and that there is a currently unexplained factor that leads to excitation of these ubiquitous fluorophores in flying squirrel fur.

There are two experiments conducted in this study: fluorescence analysis and mass spectrometry analysis. The fluorescence analysis is performed by exciting extracted samples at fixed wavelength of 350 nm and scanning emissions from 400 nm to 600 nm; no rationale is given for the choice of these parameters. A glaring weakness of the selected excitation and emission wavelengths is that these parameters are ill-suited to detect the presence of porphyrins: the compounds the authors state are “...consistent with [their] findings.” Most porphyrins associated with fluorescence in animals absorb strongest near 400-410 nm and emit between 600-750 nm. As such, it would be difficult to deduce porphyrin-based fluorescence from the parameters used. Moreover, the authors note that both fluorescence fur and some non-fluorescent fur show some emission activity, which leads to the unsubstantiated claim that “some factor currently unexplained likely leads to excitation of fluorophores in flying squirrel fur.” The authors make no attempt to prescribe a structure to this seemingly ubiquitous fluorophore, but to this reviewer, all of the fluorescence spectra look like melanins, which are expected to be present in all types of fur regardless of fluorescence.

The second experiment executed in this study is mass spectrometry. The authors run all of the fur samples under both positive and negative mode and use commercial software to delineate features from the fluorescence samples (ventral and dorsal) from non-fluorescent samples. The authors arrive at 22 features that have some association with fluorescence but make no attempt to proffer reasonable chemical structures for these features. The authors do provide software/database generated chemical formulas for the 22 features; however, many of these formulas are blatantly unreasonable (mammals do not make compounds with formulas such as C9H7F3N2, C9H8F3N3, C6H11I, CH5B2, etc.). While this reviewer understands that these formulas are not directly generated by the authors, some oversight is needed to prevent assertions that may offend readers with chemical backgrounds.

While this reviewer is intrigued by the study, the questionable experimental designs coupled with the lack of meaningful interpretation leads me to recommend this manuscript be rejected for publication to PLOS ONE.

Reviewer #2: This paper tackles fluorescence in mammals, a topic of growing interest. I found it to be exceptionally clearly written, and the methods to be strong and soundly explained. I have a number of minor comments, below and two more general comments. I wish all papers were this clearly presented! It was a pleasure to read.

GENERAL COMMENTS

(1) you find the same molecules in fluorescing and non fluorescing squirrels, and you even show that the supposedly "non fluorescing squirrels" are actually capable of some degree of fluorescence. Is it possible that these results can be explained by how much of the molecules is present in each? Maybe noticeable, measurable fluorescence is determined by amount (rather than presence or absence).

(2) Fluorescence does not create light. Fluorescence is not glowing. Needless to say, YOU do not make these classic errors in this well-written paper, but the ill-equipped reader is also not directed to address these common misconceptions. This is a stylistic request of mine which you are free to ignore, but could you add some text to explain these common misconceptions?

MINOR COMMENTS

Page 2

- missing comma after "fully understood"- give an example of aromatic molecules?

- when you say that molecules with a small HOMO-LUMOgap emits UV wavelengths, I find myself taking a pause to sort out whether in this case the molecules are fluorescing UV light-- rather than as is typically described in nature, absorbing UV light and emitting higher-wavelength light in the blue or green or even higher. Is it correct as written?

Page 3

-briefly explain what a conjugated pi-bonded system is

- in the third paragraph, can you briefly add what wavelengths of light are fluoresced for these examples? It is useful to know what the incident light color is and what the fluoresced light color is

Page4

- can you say more about the evidence that nocturnal animal can detect fluorescent pigmentation? This is a conclusion that requires that there is ample UV illumination to produce fluorescence, and that the fluoresced light (as a subset of all reflected light) is sufficient to be detected by the animal in the relevant wavelength range.

- when you say fluorescent porphyrins break down upon repeated exposure to UV, so we should expect it to be more common in nocturnal mammals, can you explain with one more sentence what you mean? that is, fluorescence is just a side effect of porphyrin chemistry, and therefore only mammals with limited exposure to UV should use lots of porphyrins?

Page 5

- the paragraph starting "fluorescence in eutherian" is an excellent paragraph that frames the problem and question really well. Perhaps this paragraph should appear at the start of the introduction or near there? I leave it up to your discretion.

Page 6

- is there a table of specimen numbers that you can reference here?

- accidental comma instead of period in G. volans

- how do you know the non-fluorescing species are non-fluorescing?- did you take the fur from the same "spot" on each animal within the dorsal and ventral pelages (in case of variation)?

Page 7

-why only scan for emissions between 400 and 600 nm?

Page 9

- Is it possible that the compounds were present in very different levels in the different species, accounting for observable versus non observable fluorescence?

- indeed as I continue reading it appears that the gray and red squirrel fur are indeed capable of a small amount of fluorescence

Page 10

- very clearly written!

- when you say that the fur does not naturally fluoresce in grey and red squirrels, do you mean at an observable level or at all? (my same point as above)

Page 11

- good explanation of the porphyrin arguments

Page 12

- my understanding is that there is very little good evidence that fluorescence actually serves a perceivable purpose in signaling. Since other readers will likely approach this paragraph with the same skepticism I do, would it be possible to either explain the evidence to one degree further of specificity or temper the claims?

Warmly,

Dakota McCoy

6. PLOS authors have the option to publish the peer review history of their article (what does this mean?). If published, this will include your full peer review and any attached files.

Reviewer #1: No

Reviewer #2: No

---

## [Author Response · Author response to Decision Letter 0]

27 Nov 2021

Reviewer 1: 

7. As a general note, manuscripts are typically written with line numbers present so it is easier for reviewers to give feedback.

R7. Line numbers have been added on this version of the manuscript to aid with continued revision. 

8. Page 1, Abstract “Some factor currently unexplained...” This sentence is unclear.

R8. This sentence has been revised, see lines 24-25. 

9. Page 2, Paragraph 2 “As the electron transitions...” The paper says longer wavelength of light, but it does not state what it is longer than. The photon that is emitted is of lower energy (i.e. longer wavelength) that the original excitation photon, but this is not clear as written in the text. Since this is a biology paper, rather than get into nuances about this point, one could simply say “As the electron transitions back to its ground state, excess energy is lost in the form of a photon.”

R9. We have added the reviewer’s suggested wording on Lines 51-52. 

10. Page 2, Paragraph 2 “Since fluorescence is not...” Replace “non-energized state” with “ground state.”

R10. We have changed “non-energized state” to “ground state” on line 54. 

11. Page 2, Paragraph 2 “Since fluorescence is not...” The phrase “transmit fluorescence wavelengths” is very awkward and formally incorrect. One would just say “fluoresce.”

R11. We have changed lines 54-55 to now say fluoresce instead of “transmit fluorescent wavelength”. 

12. Page 2-3, Paragraph 2 “Other molecules that contain...” Replace “different” with “various.”

R12. Line 58 now says “various” instead of “different. 

13. Page 3, Paragraph 1 “The fluorescence of aromatics...” This is incorrect. The LUMO is higher in energy than the HOMO, therefore you cannot have a deexcitation from the HOMO to the LUMO; its LUMO to HOMO. Please correct this sentence.

R13. This error has been corrected to now have the correct order of the HOMO-LUMO gap see Line 60-63. 

14. Page 3, Paragraph 1 “Aromatic rings with highly...” The sentence is structured such that it implies the HOMO-LUMO gap emits UV light, which is technically incorrect. The HOMO-LUMO gap dictates – to an extent – the energy of the photon that is emitted, and the energy of the emitted photon dictates the wavelength of light we observe.

R14. This has been revised to better reflect the process of fluorescence. See lines 62-65. 

15. Page 3, Paragraph 3 “In fish, there are...” Add the following citation: Park HB et al (2019) Bright green biofluorescence in sharks derives from Bromo-Kynurenine metabolism. iScience 19:1291–1336. https://doi.org/10.1016/j.isci.2019.07.019

R15. Added citation to line 80. 

16. Page 4, Paragraph 2 “Tryptophan is one of...” Tryptophan is one of the three natural/proteinogenic aromatic amino acids. There are many aromatic amino acids.

R16. Change made on line 113. 

17. Page 5, Paragraph 1 “The colour of fluorescence...” Fluorescence is not within a compound.

R17. We have changed the wording here, see line 118. 

18. Page 7, Paragraph 2 “Fluorescence analysis: Aliquots ( 2 ml) Remove space before ( and 2.

R18. Fixed incorrect spacing on line 182. 

19. Page 8, Results, Paragraph 1 “Several of these compounds...” What does this mean? What are the exact criteria being used to determine whether compounds have the necessary chemical structure to indicate capability to fluoresce?

R19. Lines 55-79 we list a general set of criteria that we expect would indicate a molecule is capable of fluorescence. This is not to say that it is impossible for molecules that don’t meet these criteria to exhibit fluorescence, but rather that molecules that are known to fluoresce typically have these characteristics. These include the presence of one or more aromatic rings, or other cyclic structure, and the presence of one ore more conjugated pi-bonds. 

20. Page 9, Paragraph 4 “Our analysis of the...” The fluorescent data looks to be consistent with eumelanin. This compound is present in keratinaceous tissue and would explain why some of the non-fluorescent specimen also exhibit emission.

R20. It is possible that the observed compound here is eumelanin, however it is beyond the scope of this study to identify specific fluorophores in the fur of non-fluorescent species. 

21. Page 11, Paragraph 2 “Our findings are consistent...” If the authors believe porphyrins could responsible for the fluorescence in these squirrel samples, then why not use excitation and emission wavelengths that can detect porphyrin-based fluorescence? The authors excite at 350 nm and detect emissions from 400-600 nm; however, biological porphyrins typically have absorbance maxima near 400 nm and emission maxima near 650 nm. The manner in which the authors conducted the fluorescence analysis would make it very difficult identify porphyrins as a fluorescent agent in this study

R21. The identification of porphyrins was beyond the scope of our study. Instead, we aimed to undertake a comparison of potential compounds across a group of species using methods similar to those recommended by Kohler et al. (2019). We agree that a follow up study targeting porphyrins specifically should use different excitation and emission wavelengths. To clarify the implications of our findings to our hypothesis, we have added information on lines 311-317 and lines 381-382. 

22. Page 13, Paragraph 1 “It is also possible...” This is easily checkable via black-light, fluorescence spectrometry, absorbance spectrometry (if porphyrin-based), etc.

R22. Because this is a novel use of mass spectrometry, we felt it was important to include that it is always possible for this type of error to occur. 

23. Table 3 A potential formula of CH5B2 is completely ludicrous, especially as the ion observed is [M+Cl]-

R23. While this chemical formula does agree with the observed m/z, we agree with the reviewer that this is not a reasonable compound to be observed in mammal fur, and we have removed it from the table. Several chemical formulas containing fluorine, have also been edited. While these chemical formulas have a slightly less mass accuracy with the observed m/z values, they are more probable to be observed in mammals. 

Reviewer 2: 

GENERAL COMMENTS

25. you find the same molecules in fluorescing and non fluorescing squirrels, and you even show that the supposedly "non fluorescing squirrels" are actually capable of some degree of fluorescence. Is it possible that these results can be explained by how much of the molecules is present in each? Maybe noticeable, measurable fluorescence is determined by amount (rather than presence or absence).

R25. Yes, this is a plausible explanation. The earlier works of Levin and Flyger (1973) suggested that the bones of the eastern fox squirrel were fluorescent because of the accumulation of porphyrins, specifically Uroporphyrin I. Uroporphyrin I is not necessarily always found in mammals because it is an isomer of Uroporphyrin III that forms in the absence of certain enzymes that stimulate proper porphyrin synthesis in most mammals. However, all mammals do have porphyrins, while not having fluorescent bones. To highlight this as a potential explanation for fluorescence see lines 296-299 and 374-377. 

26. Fluorescence does not create light. Fluorescence is not glowing. Needless to say, YOU do not make these classic errors in this well-written paper, but the ill-equipped reader is also not directed to address these common misconceptions. This is a stylistic request of mine which you are free to ignore, but could you add some text to explain these common misconceptions?

R26. We have added a short explanation on lines 47-49. 

MINOR COMMENTS

27. Page 2

- missing comma after "fully understood"- give an example of aromatic molecules?

- when you say that molecules with a small HOMO-LUMOgap emits UV wavelengths, I find myself taking a pause to sort out whether in this case the molecules are fluorescing UV light-- rather than as is typically described in nature, absorbing UV light and emitting higher-wavelength light in the blue or green or even higher. Is it correct as written?

R27. 

A) A comma has been added after fully understood. 

B) An example has been added on line 56

C) The HOMO-LUMO gap itself does not emit UV light, rather the HOMO-LUMO gap dictates the energy of the photon that is emitted. See R12 and R13. 

28. Page 3

-briefly explain what a conjugated pi-bonded system is

- in the third paragraph, can you briefly add what wavelengths of light are fluoresced for these examples? It is useful to know what the incident light color is and what the fluoresced light color is

R28. 

A) We have added a brief sentence to generally describe what a pi-bonded system is. See lines 68-69. 

B) The incident light colour of each of these fluorophores has been added where applicable and has been left more general for taxa that fluoresce across the entire, or almost entire spectrum. Where possible, we have also added emissions these species are observed under. 

29. Page4

- can you say more about the evidence that nocturnal animal can detect fluorescent pigmentation? This is a conclusion that requires that there is ample UV illumination to produce fluorescence, and that the fluoresced light (as a subset of all reflected light) is sufficient to be detected by the animal in the relevant wavelength range.

- when you say fluorescent porphyrins break down upon repeated exposure to UV, so we should expect it to be more common in nocturnal mammals, can you explain with one more sentence what you mean? that is, fluorescence is just a side effect of porphyrin chemistry, and therefore only mammals with limited exposure to UV should use lots of porphyrins?

R29. 

A) You are correct with this conclusion, the two studies we are aware of that specifically look at mammals’ ability to detect UV light suggest that the density of various photoreceptor proteins, and the absence of a yellow-eye lens that would filter UV light in diurnal species, may indicate that at least some nocturnal species can detect UV light in otherwise low-light conditions. To better explain these two studies, we have added information on lines 100-105. 

B) Our current understanding would predict that all mammals would have similar levels of porphyrins (or variation in porphyrin abundance may be independent from fluorescence to a certain degree). Because porphyrins are used in the heme production pathway, fluorescence is not the primary function of these compounds. Rather, if fluorescence is largely dictated by the accumulation of porphyrins in the fur, and UV light degrades these compounds, we should see similar levels of porphyrins in the fur of any mammal regardless of active period. However, porphyrins within diurnal mammals should degrade and only produce low, or no measurable level of fluorescence, versus nocturnal mammals that should be fluorescent. This hypothesis, presented by Toussaint et al. (2021) would suggest that all, or most, nocturnal mammals should be fluorescent. See line 106-110. 

30. Page 5

- the paragraph starting "fluorescence in eutherian" is an excellent paragraph that frames the problem and question really well. Perhaps this paragraph should appear at the start of the introduction or near there? I leave it up to your discretion.

R30. We prefer to leave this as is. 

31. Page 6

- is there a table of specimen numbers that you can reference here?

- accidental comma instead of period in G. volans

- how do you know the non-fluorescing species are non-fluorescing?- did you take the fur from the same "spot" on each animal within the dorsal and ventral pelages (in case of variation)?

R31. 

A) We have added a table with a list of all specimens examined with the appropriate number of ventral/dorsal samples taken, with the category (fluorescent fur vs not) line 152. 

B) Removed comma and added a period. 

C) All fur samples were collected from specimens that were observed to be either fluorescent (flying squirrels) or non-fluorescent (chipmunks, gray squirrels and red squirrels) under a 395nm blacklight table caption 1, lines 152. 

32. Page 7

-why only scan for emissions between 400 and 600 nm?

R32. All fluorescent samples within this study, and the original paper by Kohler et al. (2019) are excited by a blacklight at 395nm. Because the ventral portion of the flying squirrel emits a pink colour, and the dorsal portion of the flying squirrel emits a blue colour, we predict the emissions range to be somewhere between 400 and 600 nm. A justification for our emissions selection has been added to the methods here 188-190: 

33. Page 9

- Is it possible that the compounds were present in very different levels in the different species, accounting for observable versus non observable fluorescence?

- indeed as I continue reading it appears that the gray and red squirrel fur are indeed capable of a small amount of fluorescence

R33. 

A) Yes, measuring the abundance or accumulation of specific potential fluorophores is a bit beyond the scope of this study. 

B) Currently, we believe that the non-fluorescent fur observed here, and potentially *all* mammal fur may have some number of fluorophores present. However, only certain species can reliably be shown to fluoresce, when examined with a blacklight at 395 nm. See line 333 for possible explanations proposed by Toussaint et al 2021. 

34. Page 10

- very clearly written!

- when you say that the fur does not naturally fluoresce in grey and red squirrels, do you mean at an observable level or at all? (my same point as above)

R34. Yes, while both gray and red squirrels may contain some potential fluorophores, when examined with a 395 nm blacklight no gray or red squirrel specimens showed any visible level of fluorescence. Unlike flying squirrels where many museum skins and wild specimens have been documented to fluoresce under these same conditions. See R22

35. Page 11

- good explanation of the porphyrin arguments

R35. Thank you. 

36. Page 12

- my understanding is that there is very little good evidence that fluorescence actually serves a perceivable purpose in signaling. Since other readers will likely approach this paragraph with the same skepticism I do, would it be possible to either explain the evidence to one degree further of specificity or temper the claims?

R36. This is an accurate conclusion based on current evidence. While some studies have proposed potential hypothesis that could suggest fluorescence has some acting environmental pressure and is therefore somehow significant in an ecological context. To our knowledge, there is no evidence that fully supports a hypothesis that fluorescence does have an ecological role.

The following has been added to highlight that our research is but one piece in the fluorescent mammal puzzle, and more research on specific biochemical pathways and relationships between fluorescent mammals is needed to conclude if there is any functional purpose to mammalian fluorescence. See lines 352-362 and lines 381-382.

---

## [Editor Report · Decision Letter 1]

10 Jan 2022

PONE-D-21-27099R1Using mass Spectrometry to investigate fluorescent compounds in squirrel furPLOS ONE

Dear Dr. Hughes,

Thank you for submitting your manuscript to PLOS ONE. After careful consideration, we feel that it has merit but does not fully meet PLOS ONE’s publication criteria as it currently stands. Therefore, we invite you to submit a revised version of the manuscript that addresses the points raised during the review process.

I have reviewed the revised version myself and find the response to the reviewers to be adequate, but I still have some concerns about the conclusions drawn by the authors. Their results were largely inconclusive, which is fine, but they need to explain more clearly what might cause some species to fluoresce and others not. Obviously there is some difference, they just didn't detect it here, and they need to provide a better explanation of what those differences might be. Please do this well, as I will not allow any more rounds of major revision.

L16: Delete “We believe that”

L23: consider->suggest

L57: delete “well known to be”

L64: delete “,”

L65: delete “we would expect that”

L284: delete “,”

L292: see L23 above

L311: This paragraph needs work. You did not find any differences between fluorescent and non-fluorescent fur. But clearly something is causing one to fluoresce and the other not to. What is it? You may not be able to answer this conclusively, but you need to provide some potential explanations, and suggestions for how to investigate them. If, as you suggest, the fluorphore responsible for fluorescence is common in squirrel fur, why does it only fluoresce in some cases? Could it be modified in some way? Or present in greater quantities? The following paragraph discusses tryptophan, but the discussion of porphyrin is unclear. How (as suggested in the abstract) do your results support a role for porphyrin in fluorescence? Are you suggesting that porphyrin is broken down or overaccumulated in the fluorescent species you studied? Here might be a good place to mention the methodological weakness mentioned by the reviewer, i.e. that your methods were not ideal for detecting porphyrin.

Could you not detect differences in abundance of compounds? If so, you should also mention that here

We look forward to receiving your revised manuscript.

Kind regards,

Matthew Shawkey

Academic Editor

PLOS ONE
---

## [Author Response · Author response to Decision Letter 1]

1 Feb 2022

Editor comments: 

1. L16: Delete “We believe that”

R1: We have removed the wording here. 

2. L23: consider->suggest

R2: We have changed “consider” to “suggest”. 

3. L57: delete “well known to be”

R3: Removed “well known to be” 

4. L64: delete “,”

R4: Removed extra comma after non-aromatic or non-conjugated

5. L65: delete “we would expect that”

R5: Removed phrase

6. L284: delete “,”

R6: Removed unnecessary comma (now L285). 

7. L292: see L23 above

R7: We have changed this to the word “suggest” (now L293). 

8. L311: This paragraph needs work. You did not find any differences between fluorescent and non-fluorescent fur. But clearly something is causing one to fluoresce and the other not to. What is it? You may not be able to answer this conclusively, but you need to provide some potential explanations, and suggestions for how to investigate them. If, as you suggest, the fluorphore responsible for fluorescence is common in squirrel fur, why does it only fluoresce in some cases? Could it be modified in some way? Or present in greater quantities? The following paragraph discusses tryptophan, but the discussion of porphyrin is unclear. How (as suggested in the abstract) do your results support a role for porphyrin in fluorescence? Are you suggesting that porphyrin is broken down or over accumulated in the fluorescent species you studied? Here might be a good place to mention the methodological weakness mentioned by the reviewer, i.e. that your methods were not ideal for detecting porphyrin.

R8: Thank you. We have revised this paragraph as follows:

L313-333: We have further elaborated on some potential factors that may cause fluorescence in some species, but not others, and have discussed factors surrounding these hypotheses. 

L340-360: We removed some information that was redundant with the new text at L312-339 and removed the information on porphyrin-based fluorescence that we believe is outside the scope of our conclusions. 

L400-406: We have adjusted our conclusions here and provided three potential explanations for our results, including the degradation of porphyrins, and potential abnormal accumulations of porphyrins. 

L414-418: We have added that fluorescent vs non-fluorescent species may also be explained by species-specific physiology supported by the findings of Pine et al. (1985). 

9. Could you not detect differences in abundance of compounds? If so, you should also mention that here

R9: L169-170 we have added a statement to the methods to clarify that our mass spectrometer analysis can confirm presence of compounds.

---

## [Editor Report · Decision Letter 2]

9 Feb 2022

Using mass Spectrometry to investigate fluorescent compounds in squirrel fur

PONE-D-21-27099R2

Dear Dr. Hughes,

We’re pleased to inform you that your manuscript has been judged scientifically suitable for publication and will be formally accepted for publication once it meets all outstanding technical requirements.

Kind regards,

Matthew Shawkey

Academic Editor

PLOS ONE
---

## [Editor Report · Acceptance letter]

11 Feb 2022

PONE-D-21-27099R2 

Using mass spectrometry to investigate fluorescent compounds in squirrel fur. 

Dear Dr. Hughes:

I'm pleased to inform you that your manuscript has been deemed suitable for publication in PLOS ONE. Congratulations! Your manuscript is now with our production department. 

Kind regards, 

on behalf of

Dr. Matthew Shawkey 

Academic Editor

PLOS ONE